# Early Immunological Effects of Ischemia-Reperfusion Injury: No Modulation by Ischemic Preconditioning in a Randomised Crossover Trial in Healthy Humans

**DOI:** 10.3390/ijms20122877

**Published:** 2019-06-13

**Authors:** Thomas H. Lange, Marco Eijken, Carla Baan, Mikkel Steen Petersen, Bo Martin Bibby, Bente Jespersen, Bjarne K. Møller

**Affiliations:** 1Department of Clinical Immunology, Aarhus University Hospital, Palle Juul-Jensens Blvd. 99, 8200 Aarhus N, Denmark; m.eijken@clin.au.dk (M.E.); mikkel.petersen@skejby.rm.dk (M.S.P.); 2Department of Renal Diseases, Aarhus University Hospital, Palle Juul-Jensens Blvd. 99, 8200 Aarhus N, Denmark; bente.jespersen@clin.au.dk; 3Department of Infectious Diseases, Aarhus University Hospital, Palle Juul-Jensens Blvd. 99, 8200 Aarhus N, Denmark; 4Department of Internal Medicine, Sector Nephrology and Transplantation, Erasmus Medical Center, Doctor Molewaterplein 40, 3015 GD Rotterdam, The Netherlands; c.c.baan@erasmusmc.nl; 5Department of Biostatistics, University of Aarhus, Bartholins Allé 2, 8000 Aarhus C, Denmark; bibby@ph.au.dk

**Keywords:** transplantation immunology, ischemia reperfusion injury, ischemic conditioning, immune regulation, early immune response, flow cytometry

## Abstract

Ischemic preconditioning (IPC) has been protective against ischemia-reperfusion injury (IRI), but the underlying mechanism is poorly understood. We examined whether IPC modulates the early inflammatory response after IRI. Nineteen healthy males participated in a randomised crossover trial with and without IPC before IRI. IPC and IRI were performed by cuff inflation on the forearm. IPC consisted of four cycles of five minutes followed by five minutes of reperfusion. IRI consisted of twenty minutes followed by 15 min of reperfusion. Blood was collected at baseline, 0 min, 85 min and 24 h after IRI. Circulating monocytes, T-cells subsets and dendritic cells together with intracellular activation markers were quantified by flow cytometry. Luminex measured a panel of inflammation-related cytokines in plasma. IRI resulted in dynamic regulations of the measured immune cells and their intracellular activation markers, however IPC did not significantly alter these patterns. Neither IRI nor the IPC protocol significantly affected the levels of inflammatory-related cytokines. In healthy volunteers, it was not possible to detect an effect of the investigated IPC-protocol on early IRI-induced inflammatory responses. This study indicates that protective effects of IPC on IRI is not explained by direct modulation of early inflammatory events.

## 1. Introduction

Organ lesions related to ischemia-reperfusion injury (IRI) are common and pose important clinical problems. IRI is the background for, e.g., myocardial infarctions, strokes, and acute kidney injury including that seen in 30–80% of patients after deceased donor kidney transplants. Circulating immune cells such as T cells are known to influence IRI [1]. This includes T helper cell type 1 and 17 (Th1 and Th17), whereas the presence of regulatory T cells (Tregs) is associated with better outcome after IRI [1,2,3,4]. IRI is associated with a pro-inflammatory immune response [5]. Cytokines are major signalling molecules, that tightly regulate the inflammatory process in the immune system, whereas IL1, IL6, IL17, TNFα and INFγ are mainly pro-inflammatory signalling and IL10 are more anti-inflammatory signalling [6].

Ischemic preconditioning (IPC) was discovered three decades ago as a method that protected against permanent ischemia in dogs hearts [7]. Initially, the main focus of IPC was cardiac IRI prevention; however, in some studies IPC was found to protect a variety of organs [8,9,10,11,12,13,14,15]. The potential use of IPC increased when a remote protective effect of IPC was demonstrated: brief episodes of non-lethal ischemia and reperfusion in one vascular bed, tissue or organ that protected another organ against failure after ischemia [16]. This phenomenon is known as remote ischemic conditioning (rIC). rIC was later simplified in a non-invasive way to make it more clinical applicable [17].

Nevertheless, translating ischemic conditioning from experimental studies into clinical practice is still a challenge, as conflicting data have been reported. A better understanding of the mechanisms of ischemic conditioning could help explain why a beneficial effect is seen in some studies, while not convincingly demonstrated in others [18,19,20]. So far limited data is available about the effects of ischemic conditioning on the immune response [1,2,4] and more knowledge regarding the interaction with the immune system might aid our understanding of how ischemic conditioning could protect against IRI. Furthermore, if a long-lasting protective effect of ischemic conditioning on the adaptive immune system is validated, ischemic conditioning might reduce a transplant patient’s need for drug-based immunosuppression.

This study evaluated the effects of IPC on IRI by analysing circulating immune cell subsets, intracellular activation markers and inflammatory-related cytokine levels, using a randomised controlled crossover trial in healthy participants. Our IPC+IRI protocol was selected based on a study showing a beneficial effect of IPC on vascular function in a simple and relatively non-invasive setting [21].

## 2. Results

### 2.1. Adverse Events

The only reported adverse events relating to IRI or IPC were discomfort in the right arm during cuff inflation and distal redness on the right arm at cuff deflation.

### 2.2. Immune Cell Subset Population

Circulating T cells (Tfh, Th1, Th17), dendritic cells (mDC and pDC) and classical monocytes were measured by flow cytometry before (baseline) and after IRI with or without IPC (time point 0, 85 min, and 24 h). All immune cells studied showed dynamic regulations after induction of IRI in the upper arm (Figure 1A–G). In contrast, the IPC procedure did not significantly affect the concentrations or the subset distribution of the immune cells analysed.

### 2.3. Phosphorylated AKT (pAKT), Stat3 (pStat3), ERK1/2 (pERK1/2) in CD4+, CD8+ T Cells and Classical Monocytes

The intracellular activation markers pAKT, pStat3 and pERK1/2 were measured in CD4+ T cells, CD8+ T cells, and classical monocytes by flow cytometry. The levels of pAKT and pStat3 were dynamically regulated after intervening with IRI (Figure 2A–I). The IPC and non-IPC groups showed identical patterns for all stainings of phosphorylated intracellular markers and no significant difference was detected.

### 2.4. Cytokine Levels

In addition to immune cells, plasma levels of 21 inflammatory-related cytokines were measured. We found no significant effect of IRI or IPC intervention (Table 1).

## 3. Discussion

To our knowledge, this is the first randomised controlled crossover trial of early immunological effects of IPC on IRI in healthy subjects. We found that circulating immune cells and early intracellular activation markers in T cells were affected by IRI. The IPC-schedule in this study did not significantly alter this pattern. Inflammatory cytokine levels were significantly affected neither by IRI nor by IPC. Cytokines are dynamically regulated in vivo and both genetic and physiological variations play a role, which might explain why we did not find a statistically significant effect of IRI or IPC on the investigated cytokines in a cohort of 19 participants. Another explanation is the fact that in a cohort of healthy men their immunoregulatory mechanisms are intact and counteracts the provocation.

We detected dynamic fluctuations of the immune cell concentrations in peripheral blood as well as the intracellular activation markers of T cells being affected by IRI. This indicated that the IRI of the model was adequate and reliable, and that the methods applied were indeed able to detect the induced differences in immune parameters. We demonstrated that an IPC-intervention procedure that could reduce myocardial injury [22,23] and favourably affected vascular function [21,24], was not able to modify the effect of IRI on the immune system. Furthermore, we observed that the number of inflammatory immune cells tended to increase more rapidly after the IRI intervention, whereas regulatory immune cells increased more slowly and seemed to remain in circulation when the inflammatory cells had left. This pattern may be expected for healthy individuals, since a primary function of regulatory cells is to prevent “overshooting” of the inflammatory response [3,25,26,27,28,29].

Our primary interest in the immunological effect of IPC was the hypothesis that IPC could favourably affect the delicate balance of the immune system after organ transplantation. Generally, it is important to reveal whether and how the immune system is involved in the mechanism of action (MOA) of IPC, and also whether IPC could induce adverse immunological effects. Understanding the mechanism of action could help explain why ischemic conditioning is effective in some studies and ineffective in others, thereby improving the clinical assessment and utilization of ischemic conditioning treatment. We were not able to demonstrate any early effects of IPC on the cellular immune response, although the study is designed to detect even minute immunological effects. In the clinical setting, it would be more difficult to detect such effects in study patients due to possible variations in age, comorbidity, acute illness, surgery and immunosuppression and the strength of the crossover design would not be applicable.

We performed the IPC intervention with four cycles of five minutes of ischemia in healthy individuals. Our protocol was chosen based on results from Kharbanda et al. in healthy individuals showing a positive effect of IPC on vascular function [21]. However, they used three cycles of five minutes of ischemia, while we used four cycles like Botker et al. who found an increased myocardial salvage in patients undergoing rIC prior to angioplasty in acute myocardial infarction [30]. Nevertheless, we cannot exclude the possibility that our design was not ideally suited to revealing the differences between the two IPC-intervention groups: a possibility Botker et al. recently elucidated [31]. Finally, IPC may be more likely to work in critically ill patients, as some findings suggests [30,32], which could explain why we failed to rule out a clinically meaningless immunological response to IPC in healthy individuals.

Ischemic conditioning has been intensively investigated, but with varying results [13,19,22,32,33,34,35]. The exact timing of the ischemic conditioning intervention may prove essential and too much preconditioning stimulus may augment rather than attenuate the IRI [36]. The study monitored immunological effects up to 24 h after intervention, and studies in rats show that IPC can affect immunological events even later [37,38,39]. In a further study, IPC performed for seven days (but not three days or 14 days) before IRI, was associated with an increase of Tregs [39]. Our findings cannot support that a primary MOA of IPC is due to an early immunological regulatory influence. Studies of prolonged effects of IPC on immune responses in humans may be necessary to reveal whether IPC has any immunological influence. The clinical applications of IPC will be limited if IPC must be applied several hours or even days before the ischemic insults. Perconditioning, as performed in clinical studies of myocardial infarction [22] and postconditioning in an ongoing clinical trial [40] is much more applicable. Likewise, immediate preconditioning or perconditioning during transplantations or other surgical procedures with IRI-exposed organs would be logistically preferable.

The strength of our study lies in the relatively large number of healthy subjects adhering to specific standardisation requirements and the design itself - in which subjects were studied twice in a randomised crossover design. However, some limitations are also recognised. Although study investigators were blinded to the intervention performed at each visit (non-IPC or IPC), it was obviously not possible to ensure blinding of the study subjects, and the volunteers experiencing non-IPC at the first visit might be anticipating the consequences of the IPC-treatment when showing up for the second experiment.

In summary, we investigated several immune parameters using flow cytometry in a randomised controlled crossover trial with healthy individuals undergoing IRI with and without IPC. Our study verified that IRI influenced cellular immunity and when induced by IRI, cells with regulatory functions remained in peripheral blood longer than inflammatory cells. Our IPC protocol did not allow us to detect any immune modulation within 24 h after IRI, thus the protective effects seen by IPC are not likely due to an early immunological influence.

## 4. Materials and Methods

We studied the effect of IPC and IRI on the immune system and key intracellular mediators involved in the initiation of cytokine production, regulation of apoptosis, and activation of the inflammatory cells. The study was designed as a randomised, controlled crossover trial of healthy volunteers. The study was conducted in accordance with the Declaration of Helsinki, and the protocol was approved by the local ethics committee (approval date: 24 November 2015. J.no.: 1-10-72-298-15) and the Danish Data Protection Agency (approval date: 11 December 2015. J.no. 1-16-02-641-15) and registered in ClinicalTrials.gov with ID code NCT03541239 (Access date: 29 May 2018. Available online: https://clinicaltrials.gov/ct2/show/NCT03541239).

### 4.1. Participants

Twenty-one healthy male volunteers (20–72 years) were recruited. The volunteers gave written informed consent prior to their participation. All subjects were non-smokers and clinically well, received no medical treatment and had normal plasma electrolytes and no albuminuria. The participants abstained from hard physical exercise for three days, from alcohol and caffeine-containing drinks for 24 h, and each participant fasted for at least six hours before each visit. Upon arrival, participants received a standardised light meal. Room temperature was maintained at 22–26 °C.

Subjects were recruited online via www.forsoegsperson.dk (Access date: 17 March 2016), at Aarhus University Hospital, and Aarhus University.

One volunteer failed to meet inclusion criteria, and one volunteer opted out because of discomfort during ischemic conditioning. In total, nineteen subjects participated in the study.

### 4.2. Randomisation and Blinding

Each subject was block-randomised in block sizes of six to begin the study with either IPC or non-IPC. Participants arrived at 9.00 a.m. at Visit 1 and at 8.00 a.m. the following day. Two to eight weeks later, participants arrived at 9.00 a.m. at Visit 2 and at 8.00 a.m. the following day. If randomised to IPC, the IPC-procedure was performed during Visit 1. If randomised to non-IPC, the IPC-procedure was performed during Visit 2. The physician was blinded to the randomisation and remained blinded during data analysis. The participants could not be blinded.

### 4.3. IPC-Procedure and Time Plan

Participants were placed on a bed with adjustable inclination (up to 45 degrees), and a standardised light meal was served. The baseline blood sample was taken on the left arm when the patient had eaten the meal and relaxed for 30 min.

The IPC was induced by cuff inflation on the right upper arm to 200 mmHg for five minutes four times, each followed by five minutes of reperfusion. This step was the only difference between the IPC- and the non-IPC experiment, and the step was replaced by a relaxation period of the same duration for non-IPC procedures. The IRI was induced by cuff inflation on the right upper arm to 200 mmHg for twenty minutes followed by reperfusion for fifteen minutes before blood sampling. In addition to baseline blood sampling at −75 min, blood was sampled at 0, 85 min and 24 h after IRI.

### 4.4. Flow Cytometry

Lymphocyte subsets, monocytes and dendritic cells (DCs) in blood samples were characterised by flow cytometry using a Novocyte Flow Cytometer (ACEA Biosciences, Inc., San Diego, CA, USA) with 405, 488, and 640 nm lasers. The data were analysed in FlowJo© (LLC 2006-2016, Version 10.2, BD Biosciences, Franklin Lakes, NJ, USA) software. Forward-scatter height versus forward-scatter area dot plots were used to eliminate doublets, and forward-scatter versus side-scatter was used to identify mononuclear cells for further analysis. T helper (Th) cells were identified on the basis of bright stainings for CD3 and CD4 and subclassified into CXCR3+/CCR5+ (mainly expressed in Th1 cells), CCR6+ (mainly expressed in Th17 cells) (Appendix A) or CXCR5+ (mainly expressed in Tfh cells) cells (Appendix A). Tregs were identified by bright staining for CD4, CD25 and FoxP3 (Appendix A). DCs were classified by a lineage-marker dump-channel to eliminate cells with bright staining for CD19, CD14, CD56 or CD3, and cells with low levels of HLA-DR. DCs were classified as CD11c+/CD86+ (mainly mDC) or CD123+, ILT3+ (mainly pDC) (Appendix A). Classical monocytes were identified as CD14+ and CD16− (Appendix A). For intracellular measurements, the lymphocytes were divided into CD3+CD4+ and CD3+CD8+ cells (Appendix A).

We estimated the concentration of cells in each blood sample by direct volumetric cytometry.

#### 4.4.1. Fluorochromes

We used antibodies conjugated with Phycoerythrin (PE), Brilliant Violet 786 (BV786), Brilliant Violet 605 (BV605), Brilliant Violet 421 (BV421), fluorescein isothiocyanate (FITC), Phycoerythrin-Cyanine7 (PE-Cy7), allophycocyanin (APC), Alexa Fluor 647 (AF647), Alexa Fluor 488 (AF488) and peridinin chlorophyll protein (PerCP). A detailed description of antibodies and the associated fluorochromes is shown in Appendix A.

#### 4.4.2. Preparation Before Study Initiation

All monoclonal antibodies were titrated to optimise separation between positive and negative populations. The intracellular targets, phosphorylated ERK1/2, AKT and Stat3, were induced by stimulation with phorbol myristate acetate (PMA) and ionomycin to obtain sufficient labelling intensity for antibody titration, and the final antibody concentration was chosen as that providing the best separation between the unstimulated and the stimulated population. A viability marker was used to eliminate dead cells from the analysis (less than 0.5% dead cells were eliminated).

#### 4.4.3. Phosphospecific Flow Cytometry

Phosphospecific flow cytometry was performed on isolated peripheral blood mononuclear cells (PBMCs). After isolating PBMCs and adjusting the concentration to (1–2) × 10^7^/mL, 100 µL PBMC suspension was pipetted into a 5-mL BD Falcon™ tube for lymphocytes and a 5-mL polypropylene tube (Beckman Coulter, Brea, CA, USA) for monocytes, and 50 µL BV Stain Buffer added to both tubes. Lymphocytes were stained with antibodies against CD4 (BD Biosciences, Franklin Lakes, NJ, USA) and CD8 (BD Biosciences) and monocytes were stained for CD14 (BD Biosciences) and CD16 (BD Biosciences). After incubation for 30 min at room temperature in the dark, both tubes were fixed with 1 mL Fix/Perm (BD Biosciences) and incubated in the dark at 2–8 °C for 50 min. Fixed cells were washed twice at 2–8 °C and then permeabilised by addition of 1 mL freezing (−20 °C) Perm Buffer III (BD Biosciences) with 87.68% methanol. The samples were kept at −20 °C for 20 min before three washing steps at 2–8 °C and intracellular staining of both tubes with anti-phosphorylated (p)Stat3 (BD Biosciences), p38MAPK (BD Biosciences), pERK1/2 (BD Biosciences) and pAKT (BD Biosciences) antibodies. The lymphocytes were stained further with anti-CD3 (Biolegend, San Diego, CA, USA) antibody. The cells were finally incubated for 60 min in the dark at 2–8 °C. Cells were washed once more at 2–8 °C and analysed with a Novocyte Flow Cytometer (ACEA Biosciences, Inc., San Diego, CA, USA). For washing, Perm/Wash Buffer (BD Biosciences) was used except in the last washing step, where 5% bovine serum in phosphate-buffered saline (PBS) was used.

The intracellular detection of these activation markers was confirmed by single-cell imaging with an ImageStream (ImageStream^®X^ (Amnis^®^), Luminex Corporation, Austin, TX, USA) (Appendix A). The procedure above was used for the single-cell imaging with two modifications. Firstly, the PBMC concentration was (4–10) × 10^7^/mL and secondly, after all steps, the tube was fixed with 0.9% formaldehyde before analysation on the ImageStream.

#### 4.4.4. Flow Cytometry of FoxP3+ Cells (Tregs)

50 µL whole blood was pipetted into a 5-mL BD Falcon™ tube. After surface staining of CD4 (Biolegend) and CD25 (BD Biosciences), the tubes incubated for 30 min in the dark at room temperature. The cells were fixed by a fixation reagent (Beckman Coulter Life Sciences, Brea, CA, USA) for fifteen minutes at room temperature in the dark before permeabilisation with 300 µL Permeabilisation Reagent (Beckman Coulter Life Sciences) and intracellular staining of FoxP3 (Biolegend) for 60 min in the dark at room temperature. The cells were washed once with phosphate-buffered saline (PBS, BD Biosciences) and once with R3 (Beckman Coulter Life Sciences), before adding 250 µL BD FACSFlow™ and analysing samples on a Novocyte Flow Cytometer (ACEA Biosciences, Inc.).

#### 4.4.5. Flow Cytometry of Dendritic Cells, CD14+ Count and CD3+ CD4+ Cells

For staining, 100 µL whole blood was pipetted into a 5-mL BD Falcon™ tube. The relevant fluorochrome-conjugated antibodies were added to the tube before incubation in the dark for 30 min at ambient temperature. Afterwards, 1.5 mL of cold (2–8 °C) ammonium chloride solution (155 mM NH_4_Cl, 10 mM potassium-hydroxide-carbonate, 0.1 mM sodium-EDTA, Ampliqon) was added to the tube to lyse erythrocytes, and the cell suspension incubated in the dark at ambient temperature for up to 40 min. The cells were analysed on a Novocyte Flow Cytometer (ACEA Biosciences, Inc.).

During the experiment, all relevant spectral overlap compensation controls (based on Compbeads, BD) were run on a weekly basis. Flow cytometry instrument performance and stability was monitored by daily bead-based quality-control procedures (see Appendix A for each tube and its staining).

### 4.5. Cytokines

To measure cytokines, the Milliplex MAP Human High Sensitivity T Cell Magnetic Bead Panel kit (Merck Millipore) was applied and analysed on a Luminex 200^TM^ (Merck Millipore, Burlington, MA, USA) as recommended by the manufacturer.

### 4.6. Statistics

Data were log-transformed when necessary to achieve normality. We compared treatment groups (IPC or non-IPC) and time points by repeated-measures ANOVA using a mixed linear model with treatment and time and the interaction between the two as fixed effects, along with order of treatment and treatment day. Subject and day within subject were included as random effects. Model validation was performed by inspecting residuals. Data were analysed in Stata IC 14 (Metrika Consulting, Stockholm, Sweden). *p*-values below 0.05 were considered statistically significant.

## Figures and Tables

**Figure 1 ijms-20-02877-f001:**
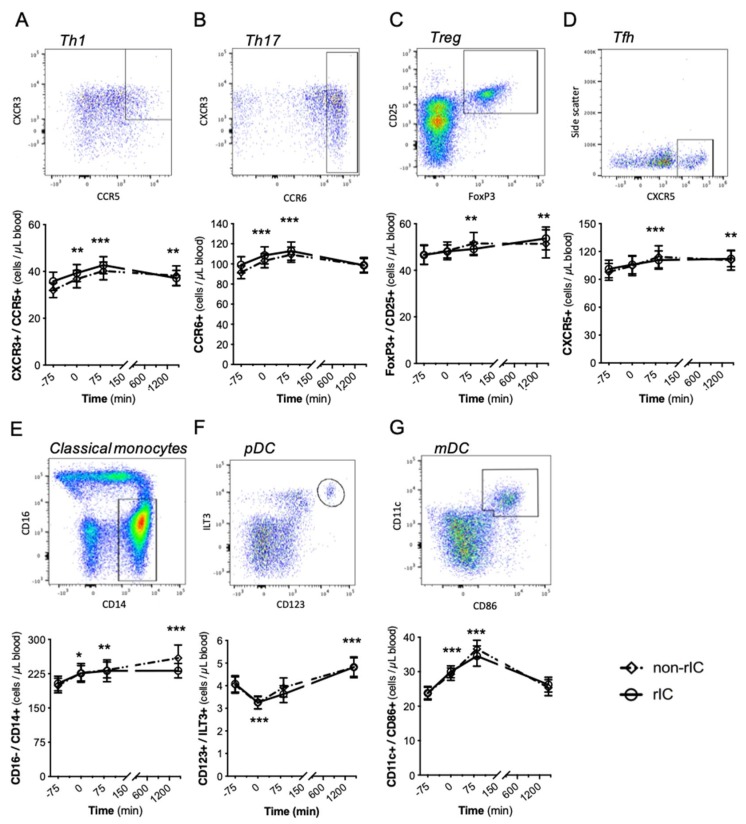
The number of immune cell subsets presented as means with 95% confidence intervals during the observation period. The *y*-axis visualises the mean level of cells/µL blood, whereas time is on the *x*-axis. No significant difference between the two groups appeared for either immune cell subsets. Dot plots for each immune cell investigated is shown above each graph. (**A**) CXCR3+/CCR5+ in Th-cells were used as marker for Th1-cells. (**B**) CCR6+ in Th-cells were used as marker for Th17-cells. (**C**) FoxP3+/CD25+ in CD4+ cells were used as markers for Tregs. (**D**) CXCR5+ in Th-cells were used as marker for Tfh. (**E**) CD16-/CD14+ were used as markers for classical monocytes. (**F**) CD123+/ILT3+ were used as markers for pDCs. (**G**) CD11c+/CD86+ cells were used as markers for mDCs. To calculate differences based on the IRI induction for the follow-up time points both the IPC and non-IPC group were combined. Statistical test: ANOVA mixed model. * *p* < 0.05, ** *p* < 0.01, *** *p* < 0.001 all compared to baseline (−75 min). IPC: Ischemic preconditioning. mDCs: Myeloid dendritic cells. pDC: Plasmacytoid dendritic cells. Tfh: T follicular helper cells. Th1 and Th17: T helper type 1 and 17 respectively. *N* = 19 healthy male participants. See Appendix A for a detailed gating strategy.

**Figure 2 ijms-20-02877-f002:**
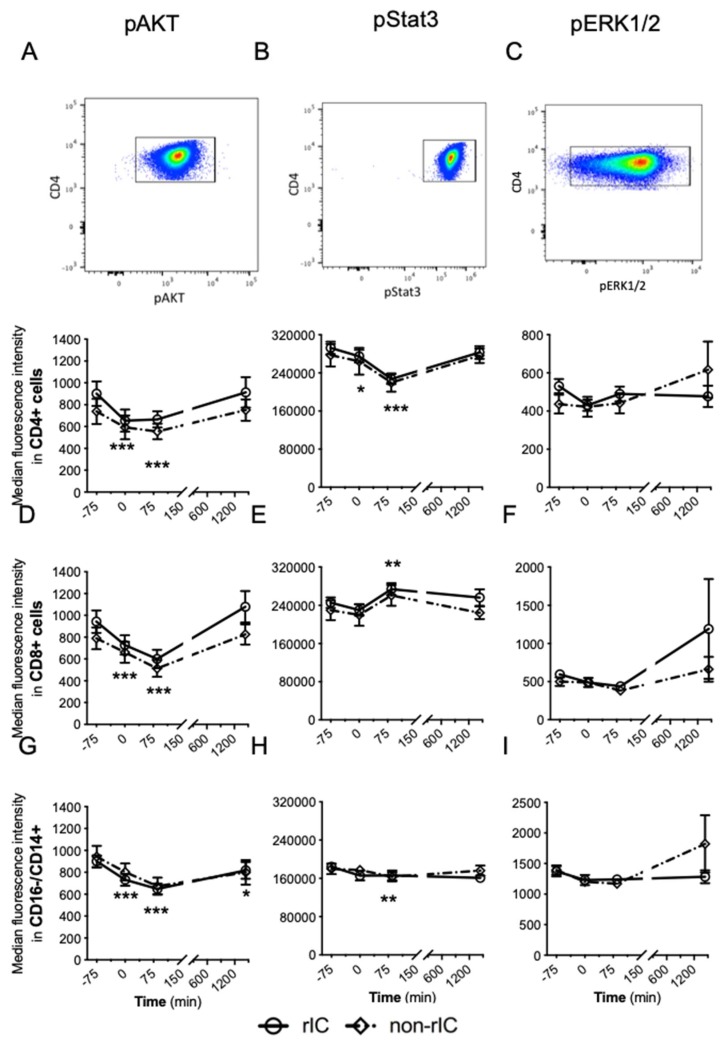
The median fluorescence intensity (MFI) of pAKT, pStat3 and pERK1/2 in CD4+ cells (**A**–**C**), CD8+ cells (**D**–**F**) and classical monocytes (**G**–**I**) with 95% confidence intervals during the observation period. The *y*-axis represents the MFI-level for the activation markers, whereas time is on the *x*-axis. Above graphs a-c a dot plot for each marker is illustrated. To calculate differences based on the IRI induction for the follow-up time points, the IPC and non-IPC group were combined. Statistical test: ANOVA mixed model. * *p* < 0.05, ** *p* < 0.01, *** *p* < 0.001 all compared to baseline (−75 min). IPC: Ischemic preconditioning. *N* = 19 healthy male participants.

**Table 1 ijms-20-02877-t001:** Shows the mean cytokine level in plasma (pg/mL) with 95% confidence intervals. “vs. Baseline” shows the difference with 95% confidence interval in the mean cytokine level between baseline and 85 min or 24 h respectively. #: Results from non-IPC and IPC are combined to see differences between baseline. Statistical test: ANOVA mixed model. IPC: Ischemic preconditioning. N.D. Not detectable. *N* = 19 healthy male participants.

	Cytokine	Baseline (mean, 95% CI) in pg/mL		85 min (mean, 95% CI) in pg/mL			24 h (mean, 95% CI) in pg/mL		
		non-IPC	IPC	non-IPC	IPC	vs. Baseline #	non-IPC	IPC	vs. Baseline #
Adaptive immunity	GMCSF	89 (−4;182)	98 (−6;202)	87 (6;168)	96 (5;188)	−2.1 (−12.7;8.5)	90 (−2;182)	89 (8;171)	−3.9 (−10.9;7.4)
	IL2	N.D.	N.D.	N.D.	N.D.	N.D.	N.D.	N.D.	N.D.
	IL4	6 (1;11)	10 (2;17)	5 (2;8)	7 (3;12)	−2.0 (−4.4;0.3)	3 (1;6)	5 (2;9)	−3.8 (−6.1;−1.4)
	IL5	1.2 (0.7;1.7)	1.4 (0.8;2.1)	1.2 (0.7;1.8)	1.4 (0.8;2.0)	−0.0 (−0.1;0.1)	1.2 (0.7;1.6)	1.3 (0.7;1.8)	−0.1 (−0.2;0.0)
	IL7	3 (2;4)	4 (3;5)	3 (2;4)	4 (3;5)	−0.1 (−0.4;0.3)	3 (2;4)	3 (2;4)	−0.2 (−0.5;0.1)
	IL13	5 (2;8)	6 (2;9)	5 (2;8)	5 (2;8)	−0.1 (−0.6;0.4)	5 (2;8)	5 (2;8)	−0.3 (−0.8;0.2)
	IL21	N.D.	N.D.	N.D.	N.D.	N.D.	N.D.	N.D.	N.D.
Pro-inflammatory signalling	ITAC	18 (13;23)	18 (14;23)	17 (13;21)	22 (16;27)	1.4 (−1.3;4.0)	18 (14;21)	18 (14;23)	−0.3 (−2.9;2.4)
	Fractalkine	70 (43;97)	84 (56;112)	73 (48;99)	76 (46;105)	−2.5 (−12.9;7.8)	75 (46;104)	68 (40;95)	−5.6 (−16.0;4.7)
	INFγ	9 (7;12)	10 (7;13)	9 (7;12)	10 (8;13)	−0.1 (−0.7;0.6)	9 (7;12)	10 (7;13)	−0.3 (−1.0;0.4)
	MIP3a	1.5 (0.4;2.5)	1.7 (0.6;2.7)	1.0 (0.3;1.7)	1.2 (0.4;2.0)	−0.5 (−1.1;0.2)	1.1 (0.3;1.9)	1.6 (0.3;2.9)	−0.2 (−0.8;0.4)
	MIP1a	12 (9;16)	13 (10;17)	12 (9;15)	13 (9;16)	−0.5 (−1.3;0.3)	12 (9;15)	12 (9;16)	−0.8 (−1.6;−0.0)
	MIP1b	7 (5;10)	8 (4;11)	7 (4;9)	8 (5;10)	−0.2 (−1.0;0.6)	6 (3;8)	7 (4;10)	−1.2 (−2.0;−0.4)
	TNFα	0.6 (0.1;1,0)	0.9 (0.2;1.6)	0.6 (−0.0;1.2)	0.8 (−0.1;1.7)	−0.0 (−0.2;0.2)	0.5 (0.0;1)	0.6 (−0.0;1.3)	−0.2 (−0.4;0.0)
	IL1b	0.6 (0.4;0.8)	0.7 (0.5;1.0)	0.6 (0.4;0.8)	0.6 (0.4;0.9)	−0.1 (−0.2;0.0)	0.6 (0.4;0.8)	0.6 (0.4;0.9)	−0.1 (−0.2;0.0)
	IL6	0.8 (0.4;1.2)	0.9 (0.5;1.3)	0.8 (0.4;1.1)	0.9 (0.6;1.3)	−0.0 (−0.2;0.1)	0.9 (0.4;1.4)	0.8 (0.4;1.2)	−0.2 (−0.2;0.1)
	IL8	1.6 (1.0;2.2)	1.7 (1.0;2.4)	1.6 (0.9;2.2)	1.8 (1.2;2.4)	0.0 (−0.1;0.1)	1.5 (0.9;2.0)	1.6 (1.0;2.3)	−0.1 (−0.3;0.0)
	IL12	1.4 (0.7;2.1)	1.7 (1.1;2.4)	1.4 (0.8;2.1)	1.7 (1.0;2.3)	−0.0 (−0.2;0.2)	1.4 (0.8;2.1)	1.5 (0.8;2.1)	−0.1 (−0.3;0.0)
	IL17a	4 (2;5)	5 (3;7)	4 (2;5)	4 (3;6)	−0.3 (−0.8;0.2)	4 (2;5)	4 (2;6)	−0.5 (−1.1;−0.0)
	IL23	946 (200;1692)	770 (81;1460)	581 (−37;1200)	768 (78;1458)	−184 (−561;194)	1304 (487;2121)	940 (193;1687)	264 (−113;641)
Anti−inflammatory signalling	IL10	N.D.	N.D.	N.D.	N.D.	N.D.	N.D.	N.D.	N.D.

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
