# Peer review of "Early Immunological Effects of Ischemia-Reperfusion Injury: No Modulation by Ischemic Preconditioning in a Randomised Crossover Trial in Healthy Humans"

_ijms, 2019, doi:10.3390/ijms20122877_

Round 1

Reviewer 1 Report

The authors present randomized controlled crossover study that evaluates the effects of IPC on IRI by analyzing circulating immune cell subsets, intracellular activation markers and inflammatory related cytokine levels in healthy volunteers.

One could argue that used IRI protocol may not be a perfect surrogate for any disease related IRI, but naturally such may be impossible to achieve in healthy subjects. Methods used for intervention are therefore suitable for this kind of analysis and all laboratory methodology is also correct.

Results of the study show that used IRI protocol was able to induce ischemia and reperfusion injury related cellular lever changes on study population and that used IPC protocol had no effect on these events.

Paper is well written and results are clearly presented. Figures and supplemental material are good. Authors have critically discussed their results and limitations and advantages of their study.

I found few minor details that should be corrected prior to publication:

Line 83: the IPC and non-IPC group were combined. *= p<0,05, **= p<0,01, ***=d p<0,001;

Two references lack details:

Ref 19. Krogstrup NV, Oltean M, Nieuwenhuijs-Moeke GJ, Dor FJ, Moldrup U, Krag SP, et al. Remote ischemic conditioning on recipients of deceased renal transplants does not improve early graft function: A multicentre randomised, controlled clinical trial. American journal of transplantation : official journal of the American Society of Transplantation and the American Society of Transplant Surgeons. 2016.

Am J Transplant. 2017 Apr;17(4):1042-1049. doi: 10.1111/ajt.14075. Epub 2016 Nov 9.

Ref 29. Bacchetta R, Barzaghi F, Roncarolo MG. From IPEX syndrome to FOXP3 mutation: a lesson on immune dysregulation. Annals of the New York Academy of Sciences. 2016

Ann N Y Acad Sci. 2018 Apr;1417(1):5-22. doi: 10.1111/nyas.13011. Epub 2016 Feb 25.

Author Response

Response to reviewer 1:

We thank the reviewer for the response to our manuscript. We acknowledge that the used protocol has limitations regarding disease related IRI, and we agree that this is unavoidable in healthy individuals.

We appreciate that the reviewer identified typographical errors and insufficient details in two references, which we have corrected in the revision:

-       At line 92 we erased “d “.

-       At line 392 Ref 19 has been corrected and marked with red.

-       At line 419 Ref 29 has been corrected and marked with red.

Reviewer 2 Report

The study should be completed with the following analysis:

·      Due to the evidence that OFR (Oxygen free radicals) plays an important role in tissue injury after ischemia reperfusion injury, OFR plasma levels should be analyzed. As complementary information it would be also very interesting to analyze NO plasma levels as an important mediator of inflammation.

·      In addition, selectin levels are also important (P-selectin, E-selectin and L-selectin) in IRI processes, and there are every day more evidences of their association with immune mediated cell recruitment to injure areas.

·      In the case of stroke research, Toll like receptors (TLRs) and IFN signaling are important in IPC-mediated protection (McDonough and Weinstein., 2016), quantitative analysis of IFN and TLRs plasma levels should be also considered in this study.

·      On another hand, Transforming growth factor (TGF-β) is released as a neuroprotectant, and at different stages, after ischemia. It would be very interesting to test quantitative TGF-beta levels in plasma.

·      Arachidonic acid and his derived metabolites are important mediators of inflammation and chemotactic agents (Francischetti et al., 2010), please make complementary analysis of arachidonic acid plasma levels.

Please correct the following mistakes in the text:

·      Line 46: whereas L1

·      Line 75: …and 24 hours) All immune cells…

·      Line 93: ..with IRI (Figure 2a-i) The IPC and…

Author Response

Response to reviewer 2:

We thank the reviewer for the constructive recommendations and agree that the analyses suggested by the reviewer would be a valuable contribution to understanding the immunological effects of IRI, but unfortunately it is not technically possible to supplement the cellular assays with stainings for selectins or TLRs that would have to be performed on fresh specimens or to assess labile factors in plasma as nitrogen oxide, superoxide ions or arachidonic acid metabolites for similar reasons.

Typographical errors as pointed out by the reviewer have been corrected.

Reviewer 3 Report

That’s an interesting randomized study aiming to investigate early immunological effects of ischemia preconditioning in ischemia reperfusion injury in healthy volunteers.  Overall, it is an interesting and timely study, however a few issues need addressing.

I find a bit surprising that authors observed changes in the frequency of lymphocyte populations in response to IRI but no changes in cytokine levels, especially in early response cytokines like IL-8, IL-6.  I feel that this is not discussed adequately.  Authors also do not comment at all why they decided to study changes in Th1, Th2, Tfh cells – specialized cells of adaptive immune system which are not necessarily relevant to the IRI injury, instead of potentially more relevant cell populations f.e. activation of neutrophils, monocytes, platelets.   Given complete lack of cytokine response to ischemic insult is it likely that chosen model was not strong enough to induce meaningful ischemic response?

Additional comments:

In methods authors state “DCs were classified as pDC (CD11c+/ILT3+) or mDC (CD123+, CD86+)”, please correct.

Page 2 – Authors state that IL-12 is mainly anti-inflammatory.  That’s against accepted thinking, what is the evidence for this statement, as IL-12 is one of the main inducers of Th1 response

Figure 1 – I suggest to change axis labels to gated populations f.e. – CXCR5+ instead of Tfh and so on and describe in the text these populations as a population containing Tfh cells ect.  Gating on chemokine receptors the way authors performed it in this manuscript is a rather rough approximation of Th subsets.  For benefit of the readers and due to the concerns raised above I also suggest that authors include representative FACS plots as a part of Figure 1, still keeping gating strategy as a supplementary figure.

Fig1, 2  - please state in the legend which statistical test was used, please state n number for each figure

Figure 2. Please show example FACS plots for pSTAT3, pAKT, pERK1/3 for one of the cell populations

Authors are advised to move cytokine data to the main manuscript.

Author Response

Point 1:

I find a bit surprising that authors observed changes in the frequency of lymphocyte populations in response to IRI but no changes in cytokine levels, especially in early response cytokines like IL-8, IL-6.  I feel that this is not discussed adequately.  Authors also do not comment at all why they decided to study changes in Th1, Th2, Tfh cells – specialized cells of adaptive immune system which are not necessarily relevant to the IRI injury, instead of potentially more relevant cell populations f.e. activation of neutrophils, monocytes, platelets.   Given complete lack of cytokine response to ischemic insult is it likely that chosen model was not strong enough to induce meaningful ischemic response?

Response 1:

We thank the reviewer for this well-founded objection. Our hypothesis originated in the question if a long-lasting effect of ischemic preconditioning on the adaptive immune response might reduce a transplant patient's need for drug-based immunosuppression. Therefore, the analysis focused on T-cells of the adaptive response. We agree that the innate immune system is more likely to be immediately affected by IRI, but for a lasting effect to be hypothesized, we would expect the specialized immune cells to be affected as well. However, we do not agree that the model is not strong enough to induce an ischemic insult, since IRI consistently induced alterations in circulating cells of the adaptive immune system. The reason that the early response cytokines are not modulated in a cohort of healthy men may originate in the fact that their immunoregulatory mechanisms are intact and counteract the provocation.

We have added some further explanation in the introduction section at line 62-64 and in the discussion section at line 142-144.

Point 2:

Page 2 – Authors state that IL-12 is mainly anti-inflammatory.  That’s against accepted thinking, what is the evidence for this statement, as IL-12 is one of the main inducers of Th1 response

Response 2:

Of course, we agree that IL-12 (and IL-23) are primarily pro-inflammatory. This has been corrected in Table 1 page 6. We refer to Akdis et al our reference no. 6 in the paper at line 361.

Point 3:

Figure 1 – I suggest to change axis labels to gated populations f.e. – CXCR5+ instead of Tfh and so on and describe in the text these populations as a population containing Tfh cells ect.  Gating on chemokine receptors the way authors performed it in this manuscript is a rather rough approximation of Th subsets.  For benefit of the readers and due to the concerns raised above I also suggest that authors include representative FACS plots as a part of Figure 1, still keeping gating strategy as a supplementary figure.

Fig1, 2  - please state in the legend which statistical test was used, please state n number for each figure

Figure 2. Please show example FACS plots for pSTAT3, pAKT, pERK1/3 for one of the cell populations

Response 3:

We agree that the figures are easier understandable if the axis labels are changed to its chemokine receptors. This has been changed in Figure 1 page 3. We have also changed line 246-247 and line 250 such that the lymphocyte subsets and dendritic cell subsets are in brackets instead of the chemokines or cell markers. We also agree that a FACS plot in the main manuscript would be preferable, which has been added in Figure 1 on page 3 and in Figure 2 page 4.

Legend in Figure 1 and Figure 2 has been changed in order to explain the FACS plot in each figure. In the legend to each figure we have now stated n for number of participants and stated what statistical test was used.

Point 4:

In methods authors state “DCs were classified as pDC (CD11c+/ILT3+) or mDC (CD123+, CD86+)”, please correct.

Authors are advised to move cytokine data to the main manuscript.

Response 4:

We thank the reviewer for the suggestions and corrections to the manuscript. We have added the cytokine table as “Table 1” in page 5 and 6. Furthermore, we have corrected the method section in line 250 to “mDC (CD11c+/CD86+) or pDC (CD123+, ILT3+)”.